# Up and Down γ-Synuclein Transcription in Dopamine Neurons Translates into Changes in Dopamine Neurotransmission and Behavioral Performance in Mice

**DOI:** 10.3390/ijms23031807

**Published:** 2022-02-04

**Authors:** Rubén Pavia-Collado, Raquel Rodríguez-Aller, Diana Alarcón-Arís, Lluís Miquel-Rio, Esther Ruiz-Bronchal, Verónica Paz, Leticia Campa, Mireia Galofré, Véronique Sgambato, Analia Bortolozzi

**Affiliations:** 1Institut d’Investigacions Biomèdiques de Barcelona (IIBB), Spanish National Research Council (CSIC), 08036 Barcelona, Spain; ruben.pavia@iibb.csic.es (R.P.-C.); dalarcar5@gmail.com (D.A.-A.); lluis.miquel@iibb.csic.es (L.M.-R.); esther.ruiz@iibb.csic.es (E.R.-B.); veronica.paz@iibb.csic.es (V.P.); leticia.campa@iibb.csic.es (L.C.); 2Institut d’Investigacions Biomèdiques August Pi i Sunyer (IDIBAPS), 08036 Barcelona, Spain; mireiagalofre@ub.edu; 3Centro de Investigación Biomédica en Red de Salud Mental (CIBERSAM), ISCIII, 28029 Madrid, Spain; 4miCure Therapeutics Ltd., Tel Aviv 6423902, Israel; 5CHU de Quebec Research Center, Axe Neurosciences, Department of Molecular Medicine, Faculty of Medicine, Université Laval, Quebec City, QC G1V 4G2, Canada; raquel.rodriguez-aller.1@ulaval.ca; 6CERVO Brain Research Centre, Quebec City, QC G1J 2G3, Canada; 7Laboratory of Stem Cells and Regenerative Medicine, Department of Biomedicine, Faculty of Medicine and Health Science, University of Barcelona, 08036 Barcelona, Spain; 8Centro de Investigación Biomédica en Red de Enfermedades Neurodegenerativas (CIBERNED), ISCIII, 28029 Madrid, Spain; 9CNRS, Institut des Sciences Cognitives Marc Jeannerod, UMR 5229, 69675 Bron, France; veronique.sgambato@inserm.fr

**Keywords:** γ-synuclein, AAV vector, antisense oligonucleotide, cognitive dysfunction, dopamine, motor deficits

## Abstract

The synuclein family consists of α-, β-, and γ-Synuclein (α-Syn, β-Syn, and γ-Syn) expressed in the neurons and concentrated in synaptic terminals. While α-Syn is at the center of interest due to its implication in the pathogenesis of Parkinson’s disease (PD) and other synucleinopathies, limited information exists on the other members. The current study aimed at investigating the biological role of γ-Syn controlling the midbrain dopamine (DA) function. We generated two different mouse models with: (i) γ-Syn overexpression induced by an adeno-associated viral vector and (ii) γ-Syn knockdown induced by a ligand-conjugated antisense oligonucleotide, in order to modify the endogenous γ-Syn transcription levels in midbrain DA neurons. The progressive overexpression of γ-Syn decreased DA neurotransmission in the nigrostriatal and mesocortical pathways. In parallel, mice evoked motor deficits in the rotarod and impaired cognitive performance as assessed by novel object recognition, passive avoidance, and Morris water maze tests. Conversely, acute γ-Syn knockdown selectively in DA neurons facilitated forebrain DA neurotransmission. Importantly, modifications in γ-Syn expression did not induce the loss of DA neurons or changes in α-Syn expression. Collectively, our data strongly suggest that DA release/re-uptake processes in the nigrostriatal and mesocortical pathways are partially dependent on substantia nigra pars compacta /ventral tegmental area (SNc/VTA) γ-Syn transcription levels, and are linked to modulation of DA transporter function, similar to α-Syn.

## 1. Introduction

Synucleins are a family of small and soluble proteins (120–140 amino acids) composed of α-, β- and γ-synuclein (α-Syn, β-Syn, and γ-Syn) which have currently only been described in vertebrates [1,2]. In the human brain, they are mainly found in the neurons and concentrated in synaptic terminals, while γ-Syn is also expressed in astrocytes [3]. Their physiological functions remain largely unknown, but these proteins play a key role in the endocytosis and exocytosis processes at the synapses maintaining the neurotransmission homeostasis, including the recycling and transport of synaptic vesicles, synaptic plasticity, or chaperone functions on release mechanisms [4,5,6,7,8,9]. All synucleins have been demonstrated to accumulate abnormally in the brain tissue from patients with synucleinopathies, including Parkinson’s disease (PD), dementia with Lewy bodies, or multiple system atrophy [10,11,12,13].

Although α-Syn is the most studied protein of this family, due to its crucial role in the pathogenesis of PD [14,15,16,17,18], the other members of the synuclein family are less characterized. These proteins, encoded by three distinct genes, share highly conserved amino-terminal regions, consisting of a varying number of 11-residue repetitive sequences, each containing a six amino acid core sequence of KTKEGV [18]. Both human α-Syn and β-Syn are 62% identical in their amino acid sequence, whereas γ-Syn shares 55% of its sequence identity with α-Syn [19]. Furthermore, α-Syn and β-Syn mRNAs are abundantly expressed in different brain areas, while γ-Syn—originally identified through its increased expression in metastatic breast cancer [20]—is specifically localized in the brainstem monoaminergic nuclei [4,21]. A body of evidence has shown that both α-Syn and γ-Syn proteins are involved in the control of dopamine (DA) homeostasis. Indeed, the enzymatic activity of tyrosine hydroxylase (TH) is regulated by direct interactions with α-Syn and γ-Syn, but not β-Syn protein [22]. Furthermore, in vitro and in vivo studies indicated that both α-Syn and γ-Syn modulate the expression and function of dopamine transporter (DAT), suggesting that DAT is negatively modulated by increases in endogenous levels of α-Syn and γ-Syn and vice versa [22,23,24,25]. Consistent with these observations, experiments using cell lines as well as cultured rat neurons have shown a dose-dependent reduction of cell surface DAT expression and function by α-Syn [24,26]. Likewise, the overexpression or knockdown of endogenous α-Syn levels in DA neurons of mice led to opposite effects on DA reuptake function by nomifensine, a DAT/norepinephrine transporter-NET inhibitor [21]. Using double α-Syn and γ-Syn knockout mice, Senior et al. [27] reported an increased DA release, supporting the hypothesis that both synucleins are involved in synaptic vesicle fusion. Moreover, mice lacking all three members of the synuclein family exhibited increased impulse-evoked DA release in the striatum accompanied by a decrease in DA tissue content [28].

Therefore, γ-Syn appears to be involved in the modulation of DA neurotransmission, although the exact mechanisms and their consequences have not yet been identified. In addition, one aspect to keep in mind is that much of the experimental evidence has been obtained using double or triple α-Syn/β-Syn/γ-Syn knockout mice, which display a prominent behavioral and neurochemical phenotype but provide limited data on the biological role of γ-Syn in the DA system. More recent studies show that single γ-Syn knockout mice display low levels of anxiety-like behavior, high exploratory activity, and enhanced habituation [29]. In addition, inactivation of the γ-Syn gene led to the improvement of working memory in mice, effects linked to changes in DA function [29,30].

To better understand the role of γ-Syn in DA function and to overcome compensatory changes and functional overlap, we have developed two adult mouse models with (i) overexpression of γ-Syn in DA neurons induced by an adeno-associated viral vector (AAV) or (ii) γ-Syn knockdown using an indatraline-conjugated antisense oligonucleotide (IN-ASO) for selective targeting to DA neurons. We revealed that up-and-down changes in endogenous γ-Syn transcription levels in midbrain DA neurons translate into opposite changes in DA neurotransmission in the forebrain regions. Moreover, we found that the increased γ-Syn levels in the midbrain DA system correlated with an impaired cognitive function, which may ultimately offer insight into the possible link between the accumulation of γ-Syn and cognitive dysfunctions in neurodegenerative pathologies.

## 2. Results

### 2.1. Mouse Model Characterization with γ-Synuclein (γ-Syn) Overexpression in Midbrain Dopamine (DA) Neurons

We first examined γ-Syn mRNA expression by unilaterally injecting AAV10-cytomegalovirus-mouse-γ-Syn (referred to as AAV10) in the substantia nigra pars compacta (SNc) and ventral tegmental area (VTA) of mice. To investigate the time-course of γ-Syn mRNA expression, mice were sacrificed at 1, 4, 8, and 16 weeks post-injection (Appendix A). Using in situ hybridization, we detected a progressive increase of endogenous γ-Syn mRNA levels in the injected side of the midbrain (SNc and VTA) compared to the control groups (vehicle- and empty vector AAV-EV-injected mice). No changes were observed on the contralateral SNc/VTA (Figure 1a,b). One-way ANOVA showed an effect of group [F_(5,23)_ = 44.75, *p* < 0.0001]. AAV10-induced γ-Syn overexpression stabilized at around 4 weeks after injection, reaching murine γ-Syn mRNA levels that slightly increased at 8 and 16 weeks (171, 185 and 193%, respectively, compared to vehicle-injected mice).

A more exhaustive histological analysis was performed only at 4 weeks post-injection in order to evaluate whether the γ-Syn mRNA co-localized with TH-positive cells, a specific DA neuronal marker. We found that around all midbrain TH-positive cells express γ-Syn mRNA (~98%), whereas AAV10 injection led to a higher density of intracellular γ-Syn mRNA in the ipsilateral SNc/VTA in comparison with the contralateral side (~3-fold, *t* = 7.218, *p* < 0.0001) (Figure 1c,d). Importantly, the number of TH-positive cells in the ipsilateral SNc/VTA was not modified over the complete rostro-caudal extent compared to contralateral SNc/VTA (Figure 1d). Likewise, α-Syn mRNA levels remained unchanged in the SNc/VTA, and a comparable γ-Syn mRNA density was detected in the raphe nuclei (RN) and locus coeruleus (LC) between the different groups of mice (Appendix A).

In addition, an increase in the number of positive cells for glutamic acid decarboxylase-GAD_67_, a specific GABA neuronal marker, expressing endogenous γ-Syn mRNA, was also detected in the ipsilateral SNc/VTA compared to contralateral SNc/VTA at 4 weeks after AAV10 injection (Figure 2a,b). Two-way ANOVA showed an effect of group [F_(1,16)_ = 12.06, *p* = 0.0031], brain hemisphere [F_(1,16)_ = 18.60, *p* = 0.0005], and interaction group-by-brain hemisphere [F_(1,16)_ = 21.72, *p* = 0.0003]. We found that GAD_67_-positive cells targeted by the AAV10 vector showed a significant augmentation in intracellular γ-Syn mRNA density in the ipsilateral SNc/VTA (*t* = 6.046, *p* = 0.0003), without changes in the number of GAD_67_-positive cells compared to the contralateral side as well as to vehicle-injected mice (Figure 2a,b). Furthermore, our data also show that the brain volume transduced by the AAV10 serotype was extensive, leaving different populations of cells outside the SNc/VTA expressing the γ-Syn gene, although we did not identify the cell types (Figure 1a). To rule out any possible effect of the viral vector itself, we performed an experiment with an AAV-EV using exactly the same protocol that we followed for the AAV10 mice. The injection of the AAV-EV did not alter the γ-Syn expression, nor did it affect behavioral performance (Appendix A).

Next, we performed immunoblot analysis of microdissected mouse SNc/VTA extracted only at 4 weeks after AAV10 injection. Western blot analysis using an antibody-recognizing murine γ-Syn showed increased γ-Syn protein levels in AAV10-injected mice compared to the vehicle-injected group (*t* = 3.650, *p* = 0.0026) (Figure 1e,f). We also used an antibody for mouse/human α-Syn, and no changes were found for α-Syn protein in the same SNc/VTA samples from AAV10- and vehicle-injected mice (Figure 1e,f).

### 2.2. Overexpression of γ-Syn in Midbrain DA Neurons Triggers Deficiencies in Forebrain DA Neurotransmission

We next performed microdialysis experiments in caudate-putamen (CPu) and medial prefrontal cortex (mPFC) of freely moving mice at 4 and 16 weeks post-injection in order to examine whether the SNc/VTA γ-Syn accumulation affects forebrain DA neurotransmission. No differences in baseline extracellular DA concentration were found in both CPu and mPFC between the different groups (Table 1), as previously also reported for α-Syn overexpression mouse model [31]. However, local veratridine infusion (50 μM, depolarizing agent) by reverse dialysis significantly increased the extracellular DA levels in CPu of vehicle-injected mice to a larger extent than in mice overexpressing γ-Syn at both 4 and 16 weeks post-injection (Figure 3a,e). Two-way ANOVA showed a marginal effect of group [F_(1,144)_ = 3.466, *p* = 0.0647], effect of time [F_(15,144)_ = 25.66, *p* < 0.0001], and interaction group-by-time [F_(15,144)_ = 4.968, *p* < 0.0001] at 4 weeks post-injection, and an effect of group [F_(1,128)_ = 14.77, *p* = 0.0002], effect of time [F_(15,128)_ = 89.53, *p* < 0.0001], and interaction group-by-time [F_(15,128)_ = 13.20, *p* < 0.0001] at 16 weeks post-injection. Similar veratridine effects on DA release were detected in mPFC of both vehicle- and AAV10-injected mice 4 and 16 weeks later (Figure 3c,g).

We have previously shown that in vivo DAT function in DA nigrostriatal and mesocortical pathways is regulated by α-Syn [21]. Here we extended this observation to assess whether changes in endogenous γ-Syn levels also affect DAT function. The infusion of DAT/NET inhibitor, nomifensine (10 and 50 μM), dose-dependently increased extracellular DA concentration in CPu and mPFC. This effect was more pronounced in the control group than in mice overexpressing γ-Syn 4 and 16 weeks later (Figure 3b,d,f,h). Two-way ANOVA showed in CPu, an effect of group [F_(1,180)_ = 58.78, *p* < 0.0001], effect of time [F_(17,180)_ = 14.15, *p* < 0.0001], and interaction group-by-time [F_(17,180)_ = 2.847, *p* = 0.0003] at 4 weeks post-injection, and an effect of group [F_(1,126)_ = 45.59, *p* < 0.0001], effect of time [F_(17,126)_ = 19.11, *p* < 0.0001], and interaction group-by-time [F_(17,126)_ = 3.069, *p* = 0.0002] at 16 weeks post-injection; in mPFC, an effect of group [F_(1,108)_ = 55.12, *p* < 0.0001], effect of time [F_(17,108)_ = 3.250, *p* < 0.0001], and interaction group-by-time [F_(17,108)_ = 2.114, *p* = 0.00110] at 4 weeks post-injection, and effect of group [F_(1,162)_ = 92.80, *p* < 0.0001], effect of time [F_(17,162)_ = 8.291, *p* < 0.0001], and interaction group-by-time [F_(17,162)_ = 3.946, *p* < 0.0001] at 16 weeks after injection. These results suggest that changes in endogenous γ-Syn levels fine-regulate DA neurotransmission, similar to α-Syn [21,22].

### 2.3. Progressive Motor and Cognitive Impairments in Mice Overexpression of γ-Syn in Midbrain DA Neurons

To investigate the potential behavioral consequences of γ-Syn overexpression in DA TH-positive neurons, we conducted a time-course study to assess changes in motor and cognitive performance. Compared to vehicle-injected mice, mice overexpressing γ-Syn in SNc/VTA exhibited poor performance of motor coordination in the rotarod at 16 weeks post-injection [F_(5,68)_ = 4.891, *p* = 0.0007], although they did not show changes in the asymmetry response as assessed in the cylinder test (Figure 4a,b). Furthermore, no alterations were found in spontaneous locomotor activity evaluated in the open field test (Figure 4c). On the other hand, mice injected with AAV-EV in SNc/VTA did not exhibit alterations in the motor behavior, suggesting that observed behavioral changes are due to midbrain γ-Syn overexpression (Appendix A).

In addition, the cognitive function was examined using different behavioral paradigms. First, mice were tested using the novel object recognition test, dependent on cortical brain areas [32,33]. During the training period, both vehicle- and AAV10-injected mice showed similar object preference, without differences between phenotypes. In contrast, AAV10-injected mice failed to discriminate between novel and familiar objects at 16 weeks post-injection [F_(5,68)_ = 14.35, *p* < 0.0001] (Figure 4d), suggesting a deficit in the short-term memory acquisition/retrieval. Next, we evaluated the latency to cross in the passive avoidance test. The initial latency was similar between vehicle- and AAV10-injected mice at 4 weeks (vehicle: 63.4 ± 19.1 s, AAV10: 78.4 ± 14.6 s) and 8 weeks post-injection (vehicle: 39.8 ± 8.77 s, AAV10: 38.9 ± 6.7 s), but significantly slower in AAV10 mice 16 weeks later compared to its respective control group (vehicle: 87.0 ± 9.54 s, AAV10: 56.93 ± 11.55 s, *p* < 0.05). The shorter retention latency in the AAV10 group at 16 weeks indicated a more severe cognitive impairment when compared to that of vehicle-injected mice [F_(5,70)_ = 6.465, *p* < 0.0001] (Figure 4e).

Moreover, we assessed the changes in spatial learning and reference memory in the Morris water maze at 4, 8, and 16 weeks post-injection. As shown in Figure 4f, task learning over the first 4 days of testing was affected in AAV10-injected mice at 8 and 16 weeks post-injection, showing significant differences in learning kinetics in comparison to vehicle-injected mice examined at the same time. Two-way ANOVA showed an effect of group [F_(3,184)_ = 13.31, *p* < 0.0001] and time [F_(3,184)_ = 6.903, *p* = 0.0002], but not interaction group-by-time. However, after removing the platform, no alterations were detected in the time spent by mice in the target quadrant, and both phenotypes preferred the target quadrant (Figure 4f).

Hence, mice overexpressing γ-Syn in SNc/VTA displayed abnormal motor coordination and balance not driven by changes in the spontaneous locomotor activity, and exhibited spatial learning and memory deficits, especially in the late phase (8 and 16 weeks post-injection), suggesting that progressive γ-Syn accumulation is directly involved in the regulation of cognitive functions.

### 2.4. In Vivo γ-Syn Knockdown Selectively in Midbrain DA Neurons Induced by IN-ASO

To gain deeper insights into the role of γ-Syn on the in vivo DA release and uptake in forebrain regions, we also generated a mouse model with a selective γ-Syn knockdown in midbrain DA neurons in SNc/VTA. Recently, we reported that IN-conjugated ASO-1233 (IN-ASO-1233) selectively reduces murine α-Syn expression in monoaminergic neurons in different animal models including wild-type mice, double mutant human A30P-A53T α-Syn transgenic mice, and nonhuman primates [31,34]. This specificity is due to the potent in vitro affinity and in vivo occupancy of IN for monoamine transporters (DAT, serotonin transporter–SERT, and NET) [35], which allows the specific accumulation of the oligonucleotide in this neuronal subset, but not in other neuronal populations, after its administration using different routes, for example intracerebral or intranasal, as previously we confirmed [21,31,34,36,37]. Here, we used a similar strategy and designed an IN-ASO-1415 sequence targeting γ-Syn mRNA in mouse midbrain DA neurons (Appendix A). Unilateral IN-ASO-1415 infusion (1 µL, 60 μg/μL) into SNc/VTA selectively reduced γ-Syn mRNA expression compared to control mice treated with vehicle or IN-conjugated nonsense ASO sequence (IN-ASO-1227) (Figure 5a,b) [F_(3,18)_ = 13.55, *p* < 0.001]. Levels of γ-Syn mRNA in SNc/VTA were significantly lower in IN-ASO-1415-treated mice than in control groups at 1 and 3 days post-administration (~25% of reduction). Furthermore, we found significant reductions in the number of TH-positive cells expressing γ-Syn mRNA, as well as reductions in the intracellular γ-Syn density in mice treated with IN-ASO-1415 (Figure 5c,d). At 1 day post-infusion, TH-positive cells showed ~28% of decrease of endogenous γ-Syn mRNA level, which remains low at 3 days after infusion (~41%) [F_(3,16)_ = 14.45, *p* < 0.0001].

Reduction of γ-Syn mRNA expression in SNc/VTA DA neurons was confirmed by Western blot procedures for murine γ-Syn protein (Figure 5e,f). We found that IN-ASO-1415 significantly decreased γ-Syn protein levels in the SNc/VTA at 3 days post-infusion (61.0 ± 2.7% versus vehicle-treated mice) [F_(2,21)_ = 7.164, *p* = 0.0042], but not at 1 day post-infusion (90.1 ± 6.3% versus vehicle-treated mice). Importantly, IN-ASO-1415 did not induce any loss of DA TH-positive cells in SNc/VTA (Figure 5d) nor changes in γ-Syn mRNA expression in RN and LC, nor in α-Syn mRNA and protein level in SNc/VTA (Figure 5e,f; Appendix A). Altogether, these data support the specificity and security of IN-ASO-1415 effects.

### 2.5. IN-ASO-Induced γ-Syn Knockdown Enhances Forebrain DA Neurotransmission

We next explored whether IN-ASO-1415-induced γ-Syn silencing in midbrain DA neurons can inversely modulate forebrain DA function compared to the AAV10-induced overexpression model. To this end, we pharmacologically modulated DA neurotransmission and performed a series of microdialysis experiments in freely moving mice at 3 days after infusion. No differences were observed in baseline DA concentrations in the CPu and mPFC among knockdown mice and control group (Table 1). Local infusion of the depolarizing agent veratridine (50 mM) by reverse dialysis significantly increased striatal DA release, with a greater effect in IN-ASO-1415-treated mice compared to control mice (Figure 6a). Two-way ANOVA showed a marginal effect of group [F_(1,112)_ = 2.98, *p* = 0.0871], effect of time [F_(15,112)_ = 551.7, *p* < 0.0001], and interaction group-by-time [F_(15,112)_ = 4.361, *p* < 0.0001]. No differences were observed in the mPFC of both phenotypes and DA release by veratrine was similar (Figure 6d).

As previously reported, synucleins are hub synaptic proteins controlling DAT expression/function ([21,22], this study). Local application of DAT/NET inhibitor, nomifensine (10–50 μM), increased dose dependently the extracellular DA concentration in the CPu and mPFC. However, unlike the γ-Syn overexpression mouse model, this effect was more pronounced in γ-Syn knockdown mice than in control group (Figure 6b,e). Two-way ANOVA showed an effect of group [F_(1,126)_ = 65.46, *p* < 0.0001], effect of time [F_(17,126)_ = 8.692, *p* < 0.0001], and interaction group-by-time [F_(17,126)_ = 3.507, *p* < 0.0001] for CPu, and an effect of group [F_(1,126)_ = 62.98, *p* < 0.0001], effect of time [F_(17,126)_ = 26.16, *p* < 0.0001], and interaction group-by-time [F_(17,126)_ = 2.441, *p* = 0.0025] for mPFC.

Moreover, in the γ-Syn knockdown mouse model, we also examined the effect of amphetamine (DA releaser and DAT inhibitor) on DA release. Striatal amphetamine application (1 and 10 μM) raised dialysate DA concentration in a dose-dependent manner, being significantly higher in IN-ASO-1415-treated mice than in the control group (Figure 6c). Two-way ANOVA showed an effect of group [F_(1,90)_ = 27.84, *p* < 0.0001], effect of time [F_(17,90)_ = 13.31, *p* < 0.0001], and interaction group-by-time [F_(17,90)_ = 2.360, *p* = 0.0048]. Likewise, local amphetamine administration (10 and 100 μM) produced a robust increase of extracellular DA concentration in the mPFC, although this effect was similar in both phenotypes (Figure 6f). Two-way ANOVA showed an effect of time [F_(17,126)_ = 32.60, *p* < 0.0001], but not effect of group nor interaction group-by-time. Like α-Syn, together this evidence confirms that γ-Syn plays a key role in the in vivo regulation of DA reuptake/release function.

## 3. Discussion

In addition to α-Syn, some evidence indicates elevated levels of γ-Syn in post-mortem brain samples from patients diagnosed with synucleinopathies, including PD, dementia with Lewy bodies, or multisystem atrophy [10,11,12,13]. Co-localization of oxidized γ-Syn and phosphorylated α-Syn has also been reported in the brain of patients with Alzheimer’s disease [12]. However, surprisingly, the biological role of γ-Syn in DA system function remains poorly understood. Indeed, understanding the biology of synuclein within DA neurons may be critical if we want to know how synuclein pathology is associated with the preferential degeneration of these neurons in PD. Here, we showed that up- and downregulation of endogenous γ-Syn transcription levels in mouse SNc/VTA significantly modifies nigrostriatal and mesocortical DA neurotransmission. Hence, DA release in CPu and mPFC was lower after γ-Syn overexpression in midbrain DA neurons, and mice progressively showed deficits mainly in cognitive functions. In contrast, acute downregulation of γ-Syn selectively in this neuronal subset led to increased DA release, largely in the nigrostriatal pathway. Despite the γ-Syn-induced fine-tuning regulation in DA neurotransmission, neither loss of DA TH-positive neurons nor alterations in α-Syn mRNA and protein levels were detected in both mouse models. Our data reveal that changes in endogenous γ-Syn transcription levels are directly involved in the regulation of DA neurotransmission. The finding that increased transcription of γ-Syn in SNc/VTA contributes to the cognitive and learning impairment observed in mice and that accumulation of γ-Syn occurs in the human brain of patients with neurodegenerative diseases [10,11,12,13] may ultimately leave γ-Syn as a potential target for therapeutic action.

Using a cytomegalovirus-driven murine γ-Syn gene AAV10 construct, we found that more than 90% of midbrain TH-positive neurons expressing murine γ-Syn mRNA reached threefold their endogenous level, leading to a remarkable increase in γ-Syn protein observed at 4 weeks after injection. Importantly, the upregulation of γ-Syn mRNA was maintained until the end of the experiment (i.e., 16 weeks after AAV10 injection). Therefore, this strategy is an appropriate mouse model to gain new insights into the function of γ-Syn in DA neurons, as it does not require extensive development and crossbreeding of γ-Syn transgenic mouse lines. However, there are some limitations despite the obvious utility of the AAV10 model reported herein. The AAV vectors represent versatile tools, both for disease modeling purposes and therapeutic uses [38,39]. When designing any given AAV-based experiment in the brain field, a number of important items regarding the AAV structure itself need to be taken into consideration, namely the selection of the best suited AAV serotype together with the choice of the most convenient promoter (e.g., the sequence driving transgene expression). It is also known that different AAV serotypes exhibit preferred cellular-specific tropism [40]. Preferential brain tropism has been reported for AAV1 to AAV10, with AAV9 and AAV10 serotypes showing the highest brain tropism [41,42]. Promoters can broadly be categorized as either ubiquitous or cell-specific, whereas a broad portfolio exists for specific expression in neurons vs. different types of glial cells, as well as within a range of neuronal phenotypes [43]. Among them, the cytomegalovirus immediate early promoter has been extensively developed and exploited for transgene expression in vitro and in vivo, including human clinical trials. The cytomegalovirus promoter has long been considered a stable, constitutive, and ubiquitous promoter for transgene expression [44,45]. In the AAV10-γ-Syn mouse model, a variable number of non-TH-positive cells located inside and outside the SNc/VTA also showed increased expression of murine γ-Syn. Indeed, several reports indicated that the SNc/VTA contains not only DA neurons but also substantial populations of GABAergic interneurons and neurons that regulate the activity of the DA neurons themselves [46,47,48,49]. Additionally, there is a small group of putative glutamatergic neurons within the VTA whose function remains unclear [50]. These glutamatergic neurons are a discrete population (i.e., they are negative for markers of DA and GABA), and are mainly located in the rostro-medial VTA [51,52]. Consistent with these observations, we found that ~30% of GABAergic GAD_67_-positive cells expressing γ-Syn mRNA in SNc/VTA showed up to a 5-fold increase in the endogenous level of γ-Syn induced by the AAV10 vector. Since GABAergic neurons play roles both in regulating the activity of DA neurons through their local collaterals axons and in regulating the activity of striatal and cortical neurons through their projected axons, we cannot exclude that AAV10-induced changes in DA neurotransmission are solely due to γ-Syn overexpression in DA neurons. In addition, other unidentified cell populations overexpressing γ-Syn mRNA in regions adjacent to the SNc/VTA should be also taken into account. Overall, the most important caveats for the AAV10-γ-Syn model are the promoter and serotype, as well as the purification and titer of AAV, as they may determine the transduction volume and spread of γ-Syn expression.

In contrast to the AAV10-γ-Syn mouse model, the IN-ASO-1415-induced γ-Syn knockdown model relied on a novel strategy, in which the oligonucleotide is covalently bound to the monoamine transporter ligands (e.g., IN herein) for selective delivery to monoamine cells, as previously demonstrated in mice and non-human primates [21,31,34,36,37]. Different IN-conjugated ASO sequences were able to selectively accumulate the oligonucleotide in mouse midbrain monoamine cells, including SNc/VTA DA neurons, but not in other non-monoamine cell populations and brain regions, after intracerebroventricular or intranasal administration [31,34]. This is due to the existence of DAT localized almost exclusively in the soma and fibers of DA neurons [53,54] for which IN has a high in vitro affinity and in vivo occupancy [35], facilitating the internalization of the oligonucleotide into DA neurons through a Rab5/Rab7-dependent endosomal mechanism [21]. We previously demonstrated that the main factor conferring the appropriate neuronal selectivity was the presence of a covalently bound ligand rather than the oligonucleotide sequence [36]. Here, we extend these observations and show that a single dose of IN-ASO-1415 (60 μg/μL, 5.2 nmol) applied locally in SNc/VTA specifically reduced γ-Syn mRNA expression (~41%) in TH-positive positive cells in parallel with a decrease in γ-Syn protein level in this brain region. In particular, three observations of the present study indicate that acute intra-SNc/VTA IN-ASO-1415 injection is safe and highly specific: (1) selective γ-Syn silencing occurred only in monoamine neurons in SNc/VTA, not in LC or RN, leaving the α-Syn mRNA unaffected; (2) it was functionally effective for at least 3 days post-injection; and (3) there were no signs of cellular toxicity in DA neurons in the injected mice.

AAV10- or IN-ASO-1415-induced up- or downregulation of γ-Syn transcription in mouse SNc/VTA led to a decrease or facilitation, respectively, of DA neurotransmission in brain projection regions, like CPu and mPFC. Changes in DA release were observed in forebrain regions after pharmacological stimulation, but not under basal conditions, suggesting that both nigrostriatal and mesocortical terminals display a standard tonic DA activity that is independent of endogenous γ-Syn transcription levels in both mouse models. Supporting this hypothesis, several reports indicated that α-Syn-null mice exhibited a normal tonic DA activity in the nigrostriatal pathway after electrical stimulation with single pulses; however, they showed increased DA release with paired stimuli that elevated Ca^2+^ levels [4]. Similarly, Senior et al. [27] showed that the DA release evoked by single action potentials in the striatum was increased only in double-null α-Syn/γ-Syn mice, but not in α-Syn-null or γ-Syn-null mice.

Previously, we reported that AAV5-induced human-α-Syn overexpression in midbrain DA neurons decreased veratridine-evoked DA release in CPu and mPFC of mice [31], and vice versa, IN-ASO-1233-induced silencing of α-Syn in DA neurons led to greater DA release in forebrain regions after local veratridine infusion than in control mice [21]. Similar results were observed after up- and downregulation of γ-Syn expression in SNc/VTA, although veratridine-evoked DA release appears to be more affected in the nigrostriatal pathway than in the mesocortical pathway in both mouse models. Recent works in PD models with overexpression of wild-type α-Syn confirmed an inhibitory effect of α-Syn on neurotransmitter release [7,31,55,56,57,58], whereas other studies come to the opposite conclusions [59,60]. Indeed, there is strong evidence showing that α-Syn has a role in the assembly of the SNARE complex, an essential process in synaptic vesicle docking and for many membrane fusion events [6,7,58]. However, to our knowledge, there are no similar studies on the effect of γ-Syn overexpression on synaptic plasticity and neurotransmitter release. What happens to the downregulation of synuclein levels? Several studies reported decreased exocytosis [61,62], no change [63], or even increased exocytosis in α-Syn-null mice [4,27,64]. Nevertheless, these studies need to be interpreted with care because α-Syn is not the only synuclein isoform, and loss of physiological function could possibly be compensated by β-Syn or γ-Syn. Studies in triple knockout mice showed an increase in synaptic transmission, and also an increase in DA release [28]. Overall, these data seem to argue for a negative regulatory role of both α-Syn, as previously reported [21,31], and γ-Syn (present study) in exocytosis processes.

In vitro and in vivo studies also demonstrated that α-Syn and γ-Syn, although the latter less well characterized, regulate the expression, trafficking and function of monoamine transporters (DAT, SERT and NET) on the cell surface [21,22,23,24,25,26,31,34]. In the present study, nomifensine (DAT/NET inhibitor) produced robust decreases or increases of extracellular DA levels in the CPu and mPFC of AAV10-γ-Syn or γ-Syn knockdown mice, respectively. Using an identical experimental protocol for nomifensine infusion, previously we showed similar results in mouse models with up and down α-Syn mRNA expression in SNc/VTA DA neurons [21,31]. Furthermore, a decrease of striatal DA reuptake with a concomitant increase of extracellular DA levels was also reported in α-Syn-null mice [65]. These findings seem to confirm that both α-Syn and γ-Syn play a key role in regulating the function of monoamine transporters (i.e., DAT) in a similar manner, consistent with a model wherein DAT is negatively modulated by increases in endogenous levels of α-Syn or γ-Syn. For α-Syn, it was proposed that it reduces DAT function by limiting cell surface expression through increased out-of-cell surface traffic, but does not appear to alter affinity for the DA substrate [22]. However, further investigation is required to address the mechanism by which γ-Syn modulates DAT function. Given the overall sequence similarity of α-Syn and γ-Syn, and the greater than 50% identity in the central region known as NAC domain [18,66], this would be a logical progression to look for a similar interaction between γ-Syn and DAT.

The effects of amphetamine (DA releaser and DAT inhibitor) on DA release in the γ-Syn knockdown mice adds more evidence demonstrating the negative modulation of DAT by γ-Syn, as previously reported for α-Syn [21]. However, the changes in extracellular DA were preferentially localized in the CPu, but not in the mPFC, of γ-Syn knockdown mice compared to the respective control mice. This finding was unexpected and suggests different mechanisms involved in the control of the active (extracellular) DA fraction in both brain areas. Previous studies indicated a lower density of DAT in mPFC compared to CPu [67,68]. Conversely, the mPFC contains a higher density of NET compared to CPu [69]. In fact, norepinephrine (NE) axons may contribute to the removal of DA from the extracellular brain space, since NET shows a similar affinity for NE and DA [70]. NET inhibitors (i.e., nomifensine, reboxetine) seem to preferentially increase the extracellular DA concentration in mPFC compared to CPu or nucleus accumbens [71,72]. Furthermore, NE axons from LC neurons may contribute to regulating extracellular DA concentration in PFC by either taking up or co-releasing DA [73,74]. Given the marked involvement of NE fibers in DA uptake in mPFC, it seems reasonable to assume that amphetamine-induced increases in DA output are greater in CPu than in mPFC, since γ-Syn expression was unchanged in LC, and, consequently, the NET function would not be altered in mice with γ-Syn knockdown in SNc/VTA. In fact, the elevation of DA output in mPFC after nomifensine confirms that NET is functionally active in IN-ASO-1415-treated mice.

Finally, recent reports showed that γ-Syn-null mice exhibit an improvement of working memory as revealed by passive and active avoidance tests [29,30]. In addition, no changes in motor functions in γ-Syn knockout mice were found [29]. On the contrary, in the current study, mice with endogenous γ-Syn overexpression showed impaired motor coordination in the rotarod test and cognitive deficits in several behavioral paradigms, especially in the last phase (8 and 16 weeks after AAV10 injection). In parallel, AAV10-γ-Syn mice showed reduced DA neurotransmission in the nigrostriatal and mesocortical pathways at the beginning of the fourth week after injection, which remained worsened until 16 weeks later. Our results clearly showed that AAV10-induced γ-Syn transcription levels were progressively increasing in SNc/VTA (increases of ~71% and 93% at 4 and 16 weeks, respectively), although they did not leave neuronal degeneration at least at 4 weeks post-injection. These previous observations emphasize the fact that the behavioral phenotype of DA deficits is not only the result of cell death, but that achieving a threshold of γ-Syn accumulation may leave functionally impaired surviving neurons that contribute to the behavioral outcome. Supporting these findings, we also previously reported that overexpression of human α-Syn in midbrain DA neurons induces cognitive dysfunctions in mice associated with a substantial loss of DA system function and a subtle effect on the number of DA neurons in SNc/VTA [31].

Overall, these findings add further insights into the cellular functions of synucleins. This work reveals the crucial contribution of γ-Syn in the regulation of DA neurotransmission that underlies motor and cognitive functions. In fact, our results indicated that DA release and re-uptake processes in the nigrostriatal and mesocortical pathways are partially dependent on γ-Syn transcription levels in SNc/VTA, and suggest that γ-Syn modulates DAT function in a similar manner to α-Syn [21,22]. The modulation of monoamine transporters by synucleins, i.e., α-Syn and γ-Syn, stands out as particularly important due to the central role of these transporters not only in normal neuronal signaling, but also in disease states which include mood disorders and neurodegenerative diseases, leaving synucleins as potential targets for therapeutic action.

## 4. Materials and Methods

### 4.1. Mice

Eight-week-old wild-type male C57BL/6J mice (*n* = 315 for the whole study, Charles River, Lyon, France) were housed under standard laboratory conditions (12 h light/dark cycle; room temperature, 23 ± 2 °C; relative humidity 50 ± 15%) with food and water available ad libitum. Animal experiments were conducted according to the 3R principles of the EU directive 2010/63/EU regarding the care and use of experimental animals and following the local laws and regulations of Generalitat de Catalunya Decree 214/97 and the University of Barcelona (UB, Barcelona, Spain) Animal Experimentation Ethics Committee (CEEA), protocol 199/20.

### 4.2. Mouse Model Overexpressing Murine γ-Syn in DA Neurons

AAV serotype 10 (AAV10) containing the murine γ-Syn cDNA under control of the cytomegalovirus (CMV) promotor (concentration: 1.34 × 10^13^ genome copies [gc]/mL) was produced by Viral Vector Production Unit (Universitat Autònoma de Barcelona-Vall d’Hebrón Institut de Recerca, Barcelona, Spain). AAV10 construct was used for the overexpression of the γ-Syn in SNc/VTA DA neurons of mice, as previously reported [31]. An empty vector (AAV-EV) with noncoding stuffer DNA was also used as a control group (concentration: 2.73 × 10^13^ gc/mL). For stereotactic delivery of AAV10, AAV-EV and vehicle (PBS-MK-Ioxidanol: 137 mM NaCl, 2.5 mM KCl, 10 mM Na_2_HPO_4_, 1.8 mM KH_2_PO_4_, 1 mM MgCl_2_ with 40% iodixanol), mice were pentobarbital-anesthetized (40 mg/kg, intraperitoneal). A volume of 1 μL of AAV10, AAV-EV or vehicle was unilaterally injected into SNc/VTA (anterior-posterior: −2.9; medial-lateral: −1.3; dorsal-ventral: −4.2 in mm) [75] using a microinjector (KDS-310-PLUS, World Precision Instruments, Sarasota, FL, USA) at 0.2 µL/min rate. The needle was kept in place for an additional 5 min before slowly being withdrawn. Topical lidocaine (10 mg/mL) and buprenorphine (0.1 mg/kg, subcutaneous) were provided as pre- and postoperative pain management. Mice were assessed at 1, 4, 8, and 16 weeks post-infusion.

### 4.3. Conjugated Antisense Oligonucleotides

The synthesis and purification of indatraline-conjugated ASO molecules targeting γ-Syn (IN-ASO-1415) or nonsense ASO sequence (IN-1227-ASO) were performed by Axolabs GmbH (Kulmbach, Germany), as previously reported [21,31,34]. ASOs are 18-mer single stranded DNA molecules with four 2′-O-methyl RNA bases at both ends to protect the internal DNA from nuclease degradation and improve the binding to the target sequence. Sequences are IN-1415-ASO [5′-ucuuCATTCTCCTCuugu-3′] and IN-ASO-1227 [5′-ccgtATCGTAAGCAgtac-3′]. In brief, ASO synthesis was performed using ultra mild-protected phosphoramidites (Glen Research, Sterling, VA, USA) and an H-8 DNA/RNA automatic synthesizer (K&A Laborgeraete GbR, Schaagheim, Germany). Indatraline hydrochloride (triple blocker of monoamine transporters) was conjugated to 5′-carboxy-C10 modified oligonucleotide through an amide bond. This condensation was carried out under organic conditions (DIPEA/DMF, 24 h). Conjugated oligonucleotides were purified by high performance liquid chromatography using an RP-C18 column (4.6 × 150 mm, 5 µm) under a linear gradient condition of acetonitrile. The molecular weight of the oligonucleotide strands was confirmed by MALDI-TOF mass spectrometry (Ultra-flex, Bruker Daktronics, Billerica, MA, USA). The concentration of conjugated sequences was calculated based on absorbance at 260 nm wavelength. Stock ASO solutions were prepared in RNAse-free water and stored at −20 °C until use.

For intra-SNc/VTA administration, mice were pentobarbital-anesthetized (40 mg/kg, intraperitoneal) and placed in the stereotaxic frame. Mice received unilaterally into SNc/VTA: IN-ASO-1415 (60 μg/μL, 1 μL), IN-ASO-1227 (60 μg/μL, 1 μL), or artificial cerebro-spinal fluid (aCSF: 125 mM NaCl, 2.5 mM KCl, 1.26 mM CaCl_2_, and 1.18 mM MgCl_2_ with 5% glucose, 1 μL) subjected to the same experimental procedure indicated above. ASO doses were chosen based on previous studies showing that were secure with no signs of neuronal and glial toxicity [21]. Mice were assessed at 24 and 72 h post-infusion.

### 4.4. In Situ Hybridization

Mice were euthanized by pentobarbital overdose and their brains were rapidly removed, frozen on dry ice and stored at −80 °C. Coronal brain sections containing SNc, VTA, RN, and LC (14 μm-thick) were obtained and processed, as described elsewhere [21,31,34]. The oligodeoxyribonucleotide probes were complementary to bases: mouse α-Syn (m-α-Syn, 411–447 sequence, GenBank accession NM_001042451), mouse γ-Syn (m-γ-Syn, 366–416 sequence, GenBank accession NM_011430), and glutamic acid decarboxylase (GAD_67_, 1600–1653 sequence, GeneBank accession NM_017007), respectively (IBIAN Technology, Zaragoza, Spain). For the α-Syn and γ-Syn assessment, each oligonucleotide was individually labeled (2 pmol) at the 3′-end with [^33^P]-dATP (>2500 Ci/mmol; DuPont-NEN) using terminal deoxynucleotidyl-transferase (TdT, Calbiochem) [21,34,35]. GAD_67_ oligonucleotide (100 pmol) was non-radioactively labeled with recombinant TdT (Roche Diagnostics GmbH, Penzberg, Germany) and digoxigenin (Dig)-11-dUTP (Boehringer Mannheim, Mannheim, Germany) according to a previously described procedure [76]. Labeled probes were purified using ProbeQuant G-50 Micro Columns (GE Healthcare, Buckinghamshire, UK).

For single or double hybridization procedure, the radioactively labeled and the non-radioactively labeled probes were diluted in a solution containing 50% formamide, 4× standard saline citrate, 1× Denhardt’s solution, 10% dextran sulfate, 1% sarkosyl, 20 mM phosphate buffer (pH 7.0), 250 μg/mL yeast tRNA, and 500 μg/mL salmon sperm DNA. The final concentrations of radioactive and Dig-labeled probes in the hybridization buffer were in the same range (~1.5 nM). Tissue sections were covered with hybridization solution containing the labeled probes, overlaid with parafilm coverslips and incubated overnight at 42 °C in humid boxes. Sections were washed four times (45 min each) in a buffer containing 0.6 M NaCl and 10 mM Tris-HCl (pH 7.5) at 60 °C, and once in the same buffer at room temperature for 30 min.

Hybridized sections were treated as previously described [21,31,34,76]. Briefly, after washing, the slides were immersed for 30 min in a buffer containing 0.1 M Tris–HCl pH 7.5, 1 M NaCl, 2 mM MgCl_2_, and 0.5% bovine serum albumin (Sigma-Aldrich, Madrid, Spain) and incubated overnight at 4 °C in the same solution with alkaline-phosphate-conjugated anti-digoxigenin-F(ab) fragments (1:5000; Roche Diagnostics GmbH, Mannheim, Germany). Afterwards, they were washed three times (10 min each) in the same buffer (without antibody), and twice in an alkaline buffer containing 0.1 M Tris–HCl pH 9.5, 0.1 M NaCl, and 5 mM MgCl_2_. Alkaline phosphatase activity was developed by incubating the sections with 3.3 mg nitroblue tetrazolium and 1.65 mg bromochloroindolyl phosphate (Roche Diagnostics GmbH, Mannheim, Germany) diluted in 10 mL of alkaline buffer. The enzymatic reaction was blocked by extensive rinsing in the alkaline buffer containing 1 mM EDTA. The sections were then briefly dipped in 70 and 100% ethanol, air-dried, and dipped into Ilford K5 nuclear emulsion (Ilford, Mobberly, Chesire, UK) diluted 1:1 with distilled water. They were exposed in the dark at 4 °C for 5–15 days (5 days for radioactive γ-Syn slides in the AAV10 model, and 15 days for radioactive γ-Syn slides in the ASO model), and finally developed in Kodak D19 (Kodak, Rochester, NY, USA) for 5 min, and fixed in Ilford Hypam fixer (Ilford, Mobberly, Chesire, UK). Tissue sections were examined with a Nikon Eclipse E100 microscope (Nikon, Tokyo, Japan). Co-expression of GAD_67_ mRNAs (Dig-labeled) with m-γ-Syn mRNA (radioactively labeled) was estimated by counting the number of Dig-labeled cellular profiles which also expressed m-γ-Syn mRNA, using a systematic random sampling tool. The number of Dig-labeled cells was considered as 100%. Only cellular profiles showing 3-fold density of silver grains higher than the background were considered to be double-labeled.

In addition, TH (anti-TH; 1:5000; ref.: AB112, Abcam, Cambridge, UK) staining was performed in nuclear emulsion-dipped m-γ-Syn hybridized slides. Briefly, following endogenous peroxidase inhibition, pre-incubation and incubation were carried out in a 1x PBS/Triton 0.2% solution containing normal serum from the secondary antibody host. Primary anti-TH was incubated for 48 h at 4 °C, followed by incubation with the corresponding biotinylated goat anti-rabbit (1:500; ref.: BA-1000, Vector Laboratories, Burlingame, CA, USA). The color reaction was performed by incubation with diaminobenzidine tetrahydrochloride solution (DAB, ref.: 5905-50TAB, Sigma-Aldrich, Madrid, Spain). Sections were mounted and embedded in Mowiol 4-88 (Sigma-Aldrich, Madrid, Spain). Microphotographs from SNc/VTA sections were acquired using a Nikon Eclipse E100 microscope (Nikon, Tokyo, Japan). The number of m-γ-Syn mRNA labelling TH-positive neurons and their intracellular density were assessed in SNc/VTA sections corresponding to different antero-posterior levels −2.46 to −3.64 mm from bregma using ImageJ (v1.51s, NIH, Bethesda, MD, USA) software. All labelled cells with their nuclei within the counting frame were counted in five consecutive sections and three different microscope fields were analyzed in each section.

For film autoradiography, some hybridized sections were exposed to Biomax-MR (Kodak, Sigma-Aldrich, Madrid, Spain) films for 24–72 h at −70 °C with intensifying screens. For specificity control, adjacent sections were incubated with an excess (50×) of unlabeled probes. Films were analyzed and relative optical densities were evaluated in three adjacent sections by duplicate of each mouse, and averaged to obtain individual values using a computer-assisted image analyzer (MCID, Mering, Germany). Figures were prepared for publication using Adobe Photoshop software (Adobe Software, San José, CA, USA). Contrast and brightness of images were the only variables we adjusted digitally.

### 4.5. Western Blot

The procedure for Western blot was performed as previously described [31]. Tissue samples of ventral midbrain containing SNc/VTA of mice were dissected from brain slices and homogenized in RIPA buffer (150 mM NaCl, 50 mM Tris pH 8.0, 1% Triton X-100, 0.5% Sodium deoxycholate, 5 mM EDTA, 0.1% SDS) with protease and phosphatase inhibitors. Proteins were quantified using PierceTM BCA Protein Assay Kit (Thermo Fisher Scientific, Waltham, MA, USA). Protein lysate (30 μg) was separated using 4–20% SDS-PAGE and electro-transferred onto a nitrocellulose membrane. Protein blots were probed with primary antibodies against anti-α-Syn (1:1000, ref.: BD610786, BD Biosciences, Franklin Lakes, NJ, USA), anti-γ-Syn (1:500, ref.: AB55424, Abcam), and anti-β-actin (1:25,000, ref.: A3854, Sigma-Aldrich, Madrid, Spain) as house-keeping control. Detection was done by chemiluminescence using SuperSignal Chemiluminescence ECL substrate kit (Thermo Fisher Scientific, Waltham, MA, USA), and pictures were taken using ChemiDoc Imaging System (Bio-Rad, Hercules, CA, USA). Images were analyzed using ImageLab software (BioRad, Hercules, CA, USA).

### 4.6. In Vivo Microdialysis

Extracellular DA concentrations were measured by in vivo microdialysis as described elsewhere [21,31,34]. One concentric dialysis probe (Cuprophan membrane; 6000 Da molecular weight cut-off; 1.5–2 mm-long) was implanted in the mouse CPu (antero-posterior 0.5, medial-lateral −1.7, and dorsal-ventral −4.5, in mm) or mPFC (antero-posterior 2.2, medial-lateral −0.2, and dorsal-ventral −3.4, in mm) [75]. Experiments were performed 48–72 h after surgery in freely moving mice. An experimenter who was blinded to mouse treatments performed microdialysis experiments. DA levels in dialysate samples were analyzed using high performance liquid chromatography (HPLC) coupled with electrochemical detection (+0.7 V, Waters 2465), with 3-fmol detection limit. The mobile phase containing 0.15 M NaH_2_PO_4_.H_2_O, 0.9 mM PICB8, 0.5 mM EDTA (pH 2.8 adjusted with orthophosphoric acid), and 10% methanol was pumped at 1 mL/min (Waters 515 HPLC pump). DA was separated on a 2.6 μm particle size C18 column (7.5 × 0.46 cm, Kinetex, Phenomenex, Torrance, CA, USA) at 28 °C.

All reagents used were of analytical grade and were obtained from Merck (Darm-stadt, Germany). DA hydrochloride and nomifensine maleate were sourced from Sigma-Aldrich-RBI. D-amphetamine sulfate and veratridine were purchased from Tocris Bioscience (Madrid, Spain). To assess the local drug effects, compounds were dissolved in aCSF (in mM: NaCl, 125; KCl, 2.5; CaCl_2_, 1.26 and MgCl_2_, 1.18) and administered by reverse dialysis at the stated concentrations (uncorrected for mem-brane recovery). Stock veratridine solution was made in DMSO and was diluted to appropriate concentrations in aCSF to reach 1% DMSO. All other drugs were dissolved in saline or aCSF, as required. Concentrated solutions (1 mM; pH adjusted to 6.5–7 with NaHCO_3_ when necessary) were stored at −80 °C, and working solutions were prepared daily by dilutions in aCSF.

### 4.7. Behavioral Testing

Behavioral analyses were performed only in the mouse model overexpressing γ-Syn in the SNc/VTA at different post-AAV10 injection times thereafter (1, 4, 8, and 16 weeks) with intervals of 1–7 days between tests. Different behavioral paradigms were used to evaluate motor and cognitive functions. All tests were performed between 10:00 and 15:00 h by an experimenter blind to mouse treatments. On the test day, mice were placed in a dimly illuminated behavioral room and were left undisturbed for at least 1 h before testing [31,34].

Cylinder test. Mice were tested for motor asymmetry in the cylinder test 1 week before surgery. Mice that present an asymmetry usage of the right-left forepaws at basal conditions were excluded from the analysis. Each mouse was placed in an acrylic cylinder (diameter, 15 cm; height, 27 cm), and the total number of left and right forepaw touches performed in 5 min was counted. Behavioral equipment was cleaned with water after each test session to avoid olfactory cues.

Rotarod. Motor coordination of mice was tested on a rotarod apparatus (Letica LE8200, Panlab, Barcelona, Spain). This consisted of a rotating rod (3 cm-diameter; hard non-slipping plastic) divided into five 5 cm lanes. On the same day of testing, mice were trained on the apparatus for three consecutive trials in which the rod was kept at constant speed (one trial at 0 rpm and two trials at 4 rpm) for 5 min. Then, mice were placed individually for 2 consecutive trials on the rod rotating at an accelerating speed from 4 to 40 rpm in 5 min. The latency at the fall of the rod and the maximal speed reached were automatically recorded.

Open field test. Motor activity was measured in four Plexiglas open field boxes 35 × 35 × 40 cm indirectly illuminated (25–40 lux) to avoid reflection and shadows. The floor of the open field boxes was covered with an interchangeable opaque plastic base that was replaced for each animal. Motor activity was recorded for 15 min by a camera connected to a computer (Video-track, Viewpoint, Lyon, France). The following variables were measured: horizontal locomotor and exploratory activity, defined as the total distance moved in cm including fast/large (speed > 10.5 cm/s) and slow/short movements (speed 3–10.5 cm/s), and the activation time.

Novelty object recognition. Mice were placed into an open field box of Plexiglas (40 × 35 × 16 cm) with a direct illumination (60 lux) in the center of the box. Two objects of identical shape were used as familiar objects. During the habituation period, the mice could freely explore the open field box without objects for 10 min on two consecutive days. In the acquisition test, the two identical objects were placed separately in the center of the open field and the mice explored it for 10 min. To minimize the presence of olfactory traces, the objects and the open field were cleaned with water between each trial. Then, 24 h after the acquisition test, one of the familiar objects was replaced by a new object and the time spent exploring of both objects (familiar and novel) was recorded for a period of 10 min. The exploration of an object was defined as pointing the nose at the object at a distance of <1 cm and/or touching it.

Passive avoidance test. The apparatus (Panlab) consisted of a rectangular box (52 × 30 × 35 cm) divided into two compartments, a highly illuminated chamber (400 lux) and a dark chamber, separated by a manually operated sliding door. On the first day, each mouse was arranged in the illuminated chamber. After 5 s of habituation period, the sliding door was opened. When the mice entered the dark chamber, the sliding door was closed, a mild foot shock was delivered (25 V, 2 s), and the mouse was left in the dark chamber for an additional 20 s. The initial latency time of entrance into the dark chamber was recorded. Initial latency times greater than 180 s were excluded from the experiment. Twenty-four hours later, each mouse was arranged in the illuminated chamber and the latency to cross was measured. The difference between both initial and test latencies was calculated to measure the retention memory time [77].

Morris water maze. An open circular pool (100 cm in diameter, 50 cm in height) was filled halfway with water and the temperature was maintained at 24 °C ± 1. Four visual clues were placed on the walls of the tank (N, E, S, and W). Non-toxic, white latex paint was added to make the water opaque, and a white escape platform was submerged 1 cm below the water level. The animals’ swimming paths were recorded by a video camera mounted above the center of the pool, and data were analyzed with SMART version 3.0 software (Panlab, Cornellà de LLobregat, Barcelona, Spain). The learning phase consisted of 4 days of trials for each mouse. The animals were submitted to four trials each day which were averaged, starting from the positions set (in random order) and without a resting phase between each trial and the subsequent one. At each trial, the mouse was placed gently into the water, facing the wall of the pool, and allowed to swim for 60 s. If not able to locate the platform in this period, the mouse was guided to the platform by the investigator. Mice were left on the platform each time for 20 s in order to allow for spatial orientation. A memory test was performed at the end of the learning days, in which the platform was removed. The escape latency during the learning tasks was measured, along with the time spent in each quadrant of the pool after the removal of the platform in the memory test.

### 4.8. Statistical Analysis

All values are expressed as the mean ± standard error of the mean (SEM). Statistical comparisons were performed using GraphPad Prism 8.01 (GraphPad software, Inc., San Diego, CA, USA) using the appropriate statistical tests, as indicated in each figure legend. Outlier values were identified by the Grubbs’ test (i.e., Extreme Student zed Deviate, ESD, method) using GraphPad Prism software and excluded from the analysis when applicable. Differences among means were analyzed by either one- or two-way analysis of variance (ANOVA) or the two-tailed Student’s *t*-test, as appropriate. When ANOVA showed significant differences, pairwise comparisons between means were subjected to Tukey’s post-hoc test or Sidak’s multiple comparisons test, as appropriate. Differences were considered significant at *p* < 0.05.

## Figures and Tables

**Figure 1 ijms-23-01807-f001:**
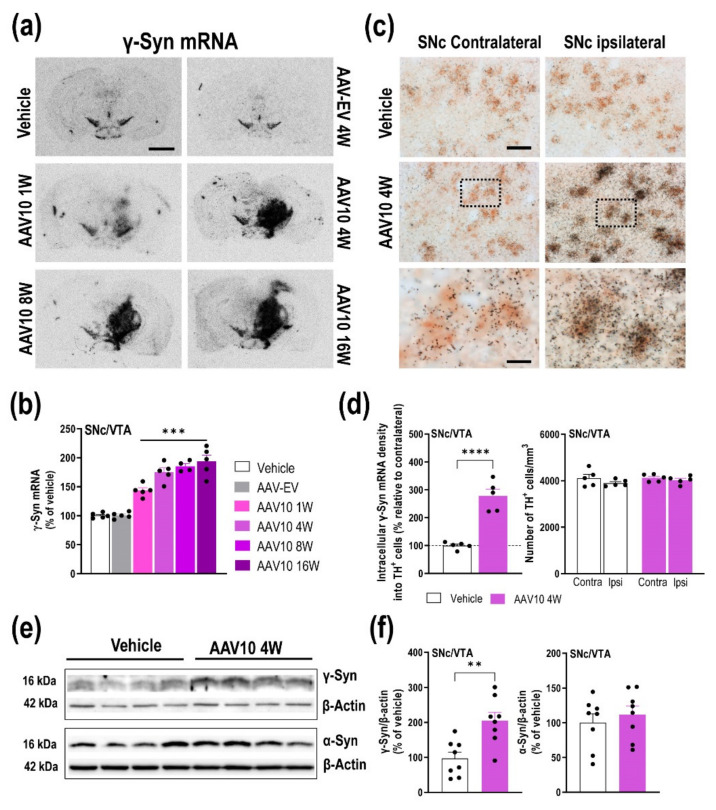
Overexpression of γ-Syn mRNA in SNc/VTA DA neurons of mice. Mice received unilaterally 1 μL of AAV10 vector containing a cytomegalovirus promoter to drive the expression of γ-Syn or vehicle into SNc/VTA and were euthanized at 1, 4, 8 and 16 weeks (W) post-injection. An empty vector (AAV-EV) with noncoding stuffer DNA was also used as a control group (see Appendix A). (**a**) Coronal brain sections showing γ-Syn mRNA levels in the SNc/VTA assessed by in situ hybridization. Scale bar: 500 μm. (**b**) Progressive increases of γ-Syn mRNA expression in the ipsilateral SNc/VTA of AAV10-injected mice compared to vehicle- and AAV-EV-injected mice (*n* = 5 mice/group). (**c**) Photomicrographs showing TH-positive cells expressing γ-Syn mRNA (^33^P-oligonucleotide silver grains) in the SNc/VTA of vehicle- and AAV10-injected mice at 4 W post-injection. Scale bar: 20 μm. Frame within the AAV10 4W image shows contralateral and ipsilateral SNc enlargements from AAV10-injected mice. Scale bar: 5 μm. Note the higher density of intracellular γ-Syn mRNA in TH-positive neurons on the ipsilateral versus contralateral side. (**d**) Dipping analyses revealed a significant increase in TH-positive cells expressing γ-Syn mRNA in the ipsilateral SNc/VTA at different antero-posterior coordinates from bregma (−2.46 to −3.64 mm). No differences in the number of TH-positive cells were found in the SNc/VTA of vehicle- and AAV10-injected mice. (**e**,**f**) Images of Western blot (**e**) and relative quantification (**f**) of murine γ-Syn and α-Syn proteins in ipsilateral SNc/VTA lysates from mice injected with vehicle or AAV10 at 4 W (*n* = 8 mice/group). β-actin was used as house-keeping control. Values are presented as mean ± SEM. ** *p* < 0.01, *** *p* < 0.001, **** *p* < 0.0001 compared to vehicle- injected mice. Abbreviations: contra, contralateral; ipsi, ipsilateral.

**Figure 2 ijms-23-01807-f002:**
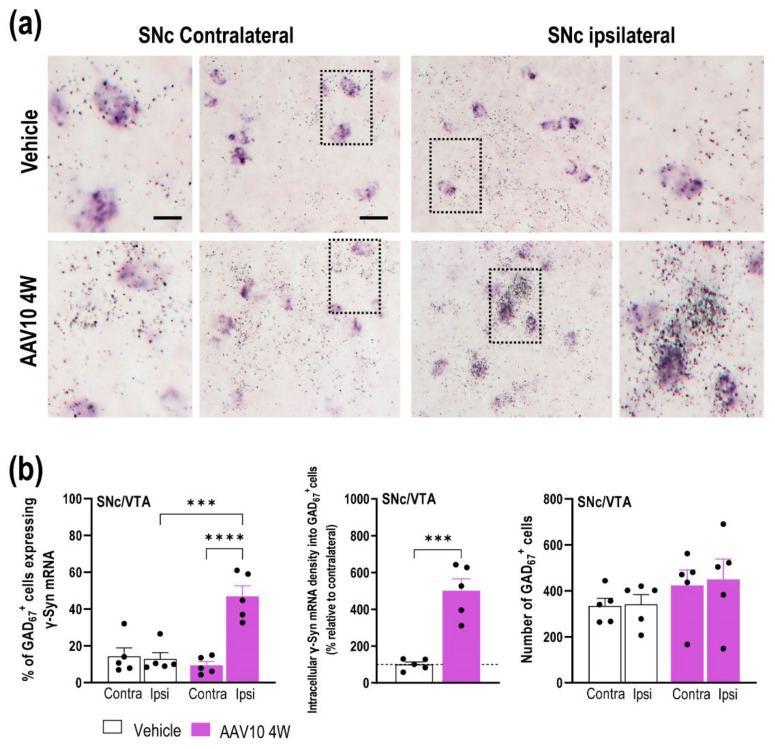
Local injection of AAV10 construct into ipsilateral SNc/VTA also induces increased γ-Syn expression in non-TH cells. Mice received unilaterally 1 μL of AAV10 vector containing a cytomegalovirus promoter to drive the expression of γ-Syn or vehicle into SNc/VTA and were euthanized at 4 weeks (W) post-injection (*n* = 5 mice/group, same mice used in Figure 1). (**a**) Photomicrographs showing GAD_67_-positive cells expressing γ-Syn mRNA (^33^P-oligonucleotide silver grains) in the SNc of vehicle- and AAV10-injected mice at 4 weeks (W) later. Scale bar: 20 μm. The frames inserted in the images show contralateral and ipsilateral SNc enlargements from vehicle- and AAV10-injected mice. Scale bar: 5 μm. Note the higher density of intracellular γ-Syn mRNA in GAD_67_-positive cells on the ipsilateral versus the contralateral side of AAV10-injected mice. (**b**) Dipping analyses revealed a significant increase in the number of GAD_67_-positive cells expressing γ-Syn mRNA in the ipsilateral SNc/VTA at different antero-posterior coordinates from bregma (−2.46 to −3.64 mm). Likewise, increases in intracellular γ-Syn mRNA density were found in GAD_67_-positive cells in the ipsilateral side of AAV10-injected mice. No differences in the number of GAD_67_-positive cells were detected in the SNc/VTA of vehicle- and AAV10-injected mice. Values are presented as mean ± SEM. *** *p* < 0.001, **** *p* < 0.0001 compared to vehicle- injected mice. Abbreviations: contra, contralateral; ipsi, ipsilateral. Black circles represent the number of mice used for each group.

**Figure 3 ijms-23-01807-f003:**
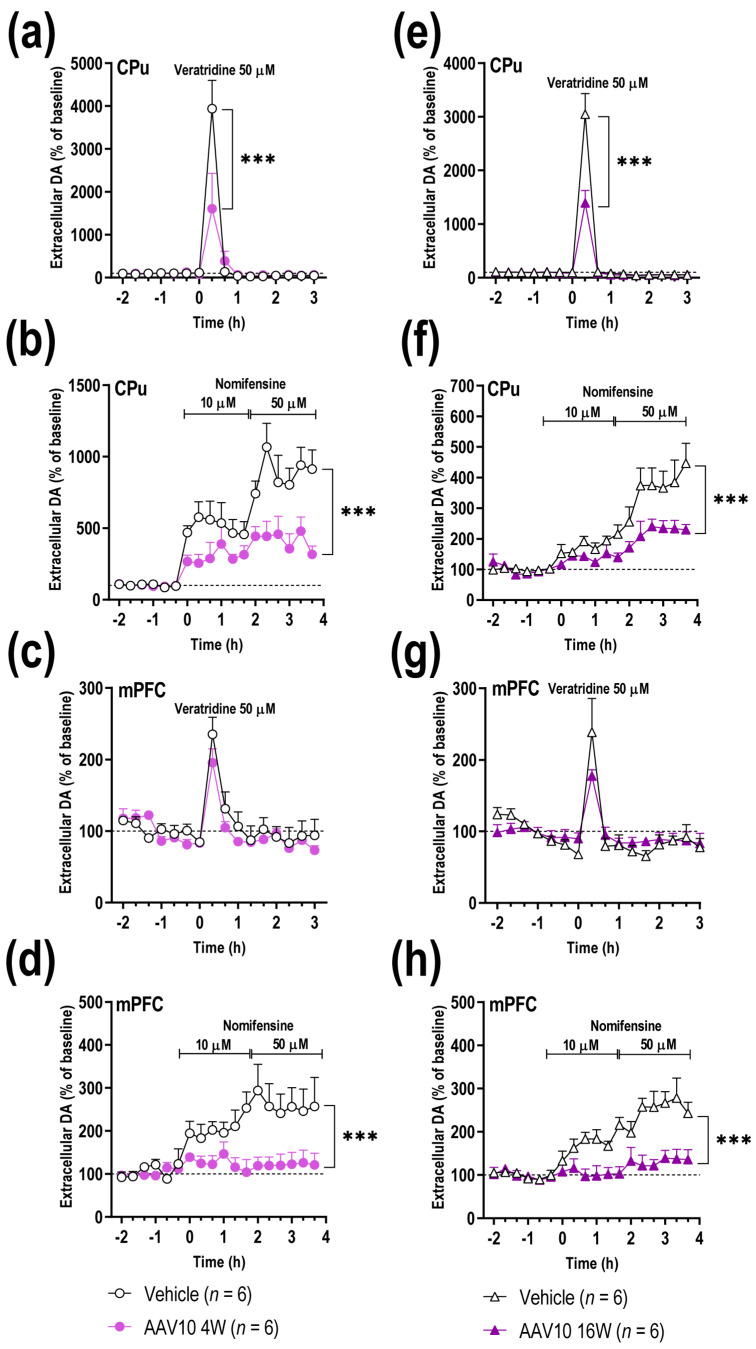
Overexpression of γ-Syn in the SNc/VTA alters forebrain DA neurotransmission. Mice received unilaterally 1 μL of AAV10 vector containing a cytomegalovirus promoter to drive the expression of γ-Syn or vehicle into SNc/VTA and were examined at 4 or 16 weeks (W) post-injection (*n* = 6 mice/group). At the end of the microdialysis procedures, the mice were euthanized and γ-Syn expression was confirmed. (**a**,**e**) Local veratridine infusion (depolarizing agent, 50 μM) by reverse dialysis increased extracellular DA release in CPu both at 4 and 16 W later. This effect was more pronounced in vehicle-injected than in AAV10-injected mice. (**b**,**f**) Nomifensine (DAT/NET inhibitor, 10 and 50 μM) increased more significantly extracellular DA levels in CPu of control group than in AAV10-injected mice. (**c**,**g**) Unlike CPu, local infusion of veratridine into mPFC augmented DA release, but this effect was similar in both phenotypes. (**d**,**h**) Local infusion of nomifensine in mPFC produced a greater effect on extracellular DA levels in vehicle-treated mice than in mice injected with AAV10 at both 4 and 16 W post-injection. Values are presented as mean ± SEM. *** *p* < 0.001, compared to vehicle-injected mice. Abbreviations: CPu, caudate-putamen; mPFC, medial prefrontal cortex.

**Figure 4 ijms-23-01807-f004:**
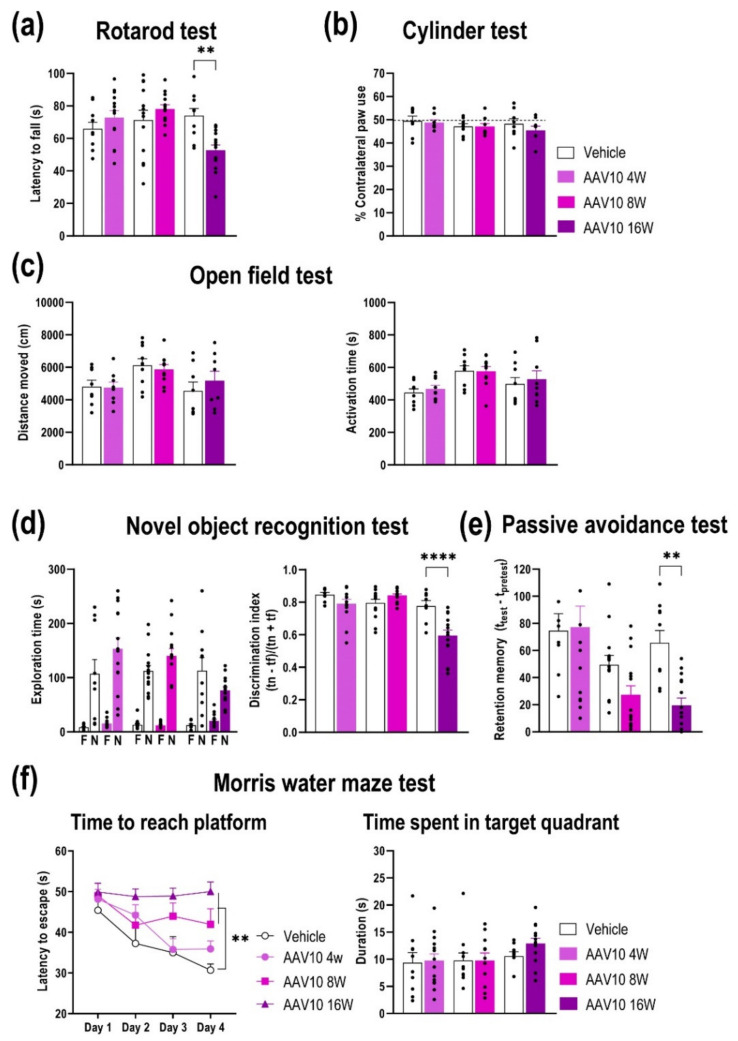
Overexpression of γ-Syn in the SNc/VTA impairs motor and cognitive behaviors. Mice received unilaterally 1 μL of AAV10 vector containing a cytomegalovirus promoter to drive the expression of γ-Syn or vehicle into SNc/VTA and were examined at 4, 8, or 16 weeks (W) post-injection (*n* = 8–12 mice/group). All mice were examined in the different behavioral paradigms. (**a**) Mice overexpressing γ-Syn showed alterations in the motor coordination in the rotarod, evaluated by a shorter latency to fall at 16 W post-AAV10 injection compared to vehicle-injected mice. (**b**,**c**) No differences were observed in the behavioral performance between the different groups in the cylinder test (**b**) as well as in the open field test (**c**). (**d**) Mice overexpressing γ-Syn failed to discriminate between familiar and novel objects (F and N, respectively) compared to vehicle-treated mice at 16 W post-injection. (**e**) Similarly, mice overexpressing γ-Syn also showed a cognitive deterioration with a shorter latency to cross in the passive avoidance test at 16 W post-injection compared to vehicle-injected mice. (**f**) In the Morris water maze, mice with increased γ-Syn expression in the SNc/VTA evoked greater latencies to reach the platform than vehicle-injected mice starting at 8 W after injection. However, no differences in time elapsed in the target quadrant were detected between the different groups of mice. Values are presented as mean ± SEM. ** *p* < 0.01, **** *p* < 0.0001, compared to vehicle- injected mice.

**Figure 5 ijms-23-01807-f005:**
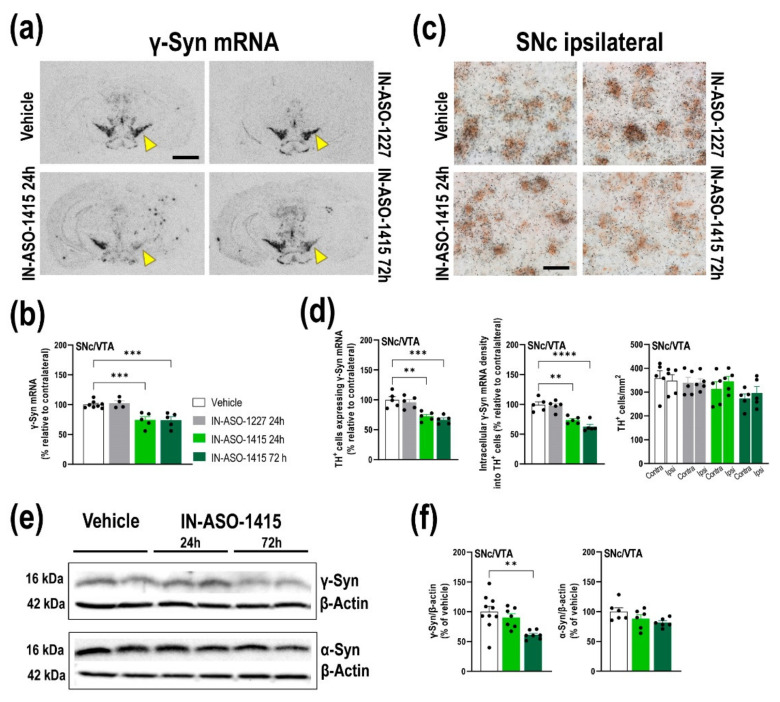
Single acute administration of IN-ASO-1415 into SNc/VTA downregulates γ-Syn expression in DA neurons. Mice received unilaterally 1 μL of IN-ASO-1415 (total dose of 60 μg) or vehicle into SNc/VTA and were euthanized at 24 or 72 h post-infusion. An IN-conjugated nonsense ASO sequence (IN-ASO-1227) was also used as a control group. (**a**) Coronal brain sections showing γ-Syn mRNA levels in the SNc/VTA assessed by in situ hybridization. Yellow arrowheads shown γ-Syn mRNA expression in SNc/VTA. Scale bar: 500 μm. (**b**) Significant reductions of γ-Syn mRNA expression in the ipsilateral SNc/VTA of IN-ASO-1415-injected mice compared to vehicle- and IN-ASO-1227-injected mice (*n* = 4–5 mice/group). (**c**) Photomicrographs showing TH-positive cells expressing γ-Syn mRNA (^33^P-oligonucleotide silver grains) in the ipsilateral SNc from mice injected with vehicle or IN-ASO sequences. Scale bar: 20 μm. Note the lower density of intracellular γ-Syn mRNA in SNc TH-positive neurons in mice that received IN-ASO-1415 than in those that received vehicle. (**d**) Dipping analyses revealed significant reductions in TH-positive cells expressing γ-Syn mRNA in the ipsilateral SNc/VTA at different antero-posterior coordinates from bregma (−2.46 to −3.64 mm). Likewise, decreases of intracellular γ-Syn mRNA density were found in TH-positive cells in the ipsilateral side of IN-ASO-1415-injected mice. No differences in the number of TH-positive cells were detected in the SNc/VTA of vehicle-, IN-ASO-1227-, and IN-ASO-1415-injected mice. (**e**,**f**) Images of Western blot (**e**) and relative quantification (**f**) of murine γ-Syn and α-Syn proteins in ipsilateral SNc/VTA lysates from mice injected with vehicle or IN-ASO-1415 at 24 and 72 h post-injection (*n* = 6–9 mice/group). β-actin was used as house-keeping control. Values are presented as mean ± SEM. ** *p* < 0.01, *** *p* < 0.001, **** *p* < 0.0001 compared to vehicle-injected mice. Abbreviations: contra, contralateral; ipsi, ipsilateral. (See Appendix A).

**Figure 6 ijms-23-01807-f006:**
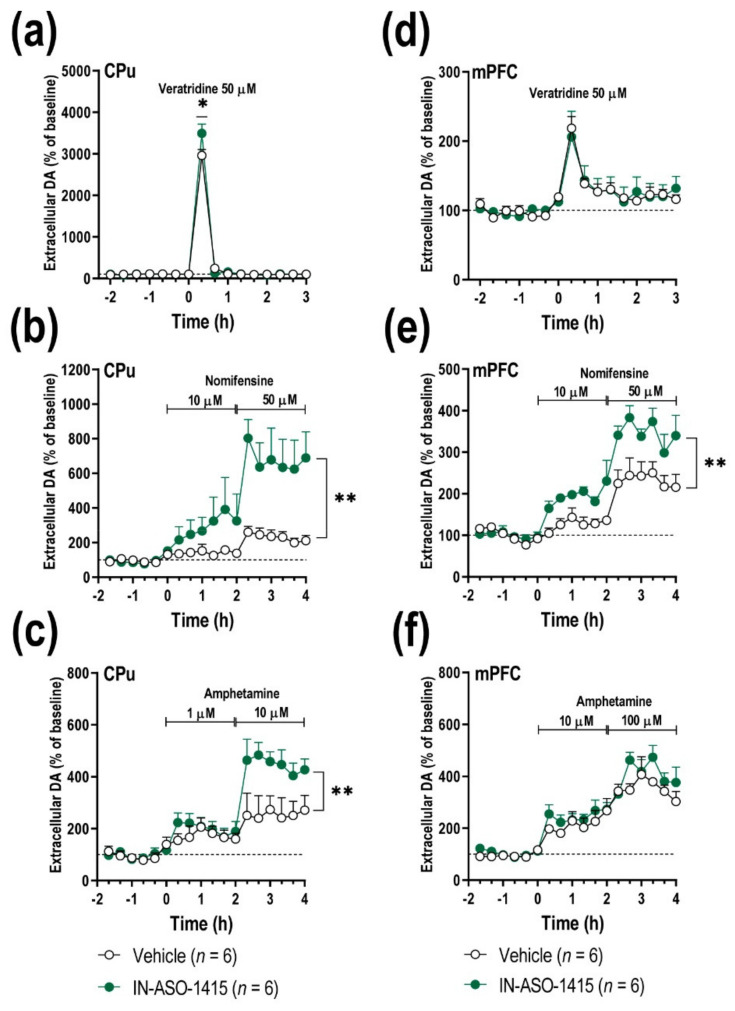
Neurochemical effects on forebrain DA neurotransmission of IN-ASO-1415-induced γ- Syn knockdown. Mice received unilaterally 1 μL of IN-ASO-1415 (total dose of 60 μg) or vehicle into SNc/VTA. Microdialysis experiments were conducted 3 days after administration (*n* = 6 mice/group). (**a**,**d**) Local veratridine infusion (depolarizing agent, 50 mM) by reverse dialysis increased extracellular DA release in CPu (**a**) and mPFC (**d**). This effect was significantly higher in the CPu of IN-ASO-1415-treated mice, but not in mPFC, compared to vehicle-treated mice. (**b**,**e**) Nomifensine (DAT/NET inhibitor, 10 and 50 μM) markedly increased extracellular DA levels in CPu (**b**) and mPFC (**e**) of γ-Syn knockdown mice than in control groups. (**c**,**f**) Similarly, local infusion of amphetamine (DA releaser and DAT inhibitor, 1–10–100 μM) induced a higher DA release in CPu (**c**), but not in mPFC (**f**) of IN-ASO-1415-treated mice versus control mice. Values are presented as mean ± SEM. * *p* < 0.05, ** *p* < 0.01, compared to vehicle- injected mice. Abbreviations: CPu, caudate-putamen; mPFC, medial prefrontal cortex.

**Table 1 ijms-23-01807-t001:** Baseline DA dialysate concentrations in CPu and mPFC of mice.

Mice	Experimental Conditions	Baseline DA	
		CPu	mPFC
Vehicle 4 weeks	aCSF aCSF + DMSO	8.1 ± 2.3 15.4 ± 2.6	18.2 ± 1.4 25.4 ± 2.9
AAV10 4 weeks	aCSF aCSF + DMSO	9.8 ± 1.1 12.8 ± 1.9	21.6 ± 3.3 20.2 ± 2.1
Vehicle 16 weeks	aCSF aCSF + DMSO	7.2 ± 1.4 7.4 ± 1.5	16.7 ± 2.8 26.2 ± 3.1
AAV10 16 weeks	aCSF aCSF + DMSO	7.4 ± 1.9 7.5 ± 1.3	13.6 ± 1.9 23.5 ± 0.6
Vehicle 3 days	aCSF aCSF + DMSO	16.2 ± 1.18 20.1 ± 1.9	19.2 ± 1.9 14.1 ± 0.7
IN-ASO-1415 3 days	aCSF aCSF + DMSO	17.6 ± 1.8 21.3 ± 4.1	20.1 ± 3.7 15.2 ± 1.9

Extracellular DA levels are expressed as fmol/20-min fraction. In the experiments involving the evaluation of the veratridine effect on extracellular DA levels, dimethyl-sulfoxide (DMSO) was added to the artificial cerebrospinal fluid (aCSF). Data are the mean ± SEM (*n* = 6 mice/group). Caudate-putamen (CPu); Medial prefrontal cortex (mPFC).

## Data Availability

Not applicable.

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
