# Peer review of "Up and Down γ-Synuclein Transcription in Dopamine Neurons Translates into Changes in Dopamine Neurotransmission and Behavioral Performance in Mice"

_ijms, 2022, doi:10.3390/ijms23031807_

Round 1

Reviewer 1 Report

The authors proposed a well-presented paper dealing with the role of γ-synuclein in dopaminergic neurons in mouse models and its consequence once accumulated on behavior and neurotransmission.

All the data have been clearly exposed and analyzed in depth, the introduction reflects the main goals of the study and conclusions discuss well the overall data inn regards to the existing literature. Cutted blots within the manuscript are completed by full blots in supplementary figures. Methods have been described clearly.

I cannot but accept this paper for publication since I've no major comments to formulate.

Author Response

Comments and Suggestions for Authors

The authors proposed a well-presented paper dealing with the role of γ-synuclein in dopaminergic neurons in mouse models and its consequence once accumulated on behavior and neurotransmission.

All the data have been clearly exposed and analyzed in depth, the introduction reflects the main goals of the study and conclusions discuss well the overall data in regards to the existing literature. Cutted blots within the manuscript are completed by full blots in supplementary figures. Methods have been described clearly.

I cannot but accept this paper for publication since I've no major comments to formulate.

Author's Reply to the Review #1

We are grateful for your time and efforts reviewing our manuscript "Up-and-down γ-synuclein transcription in dopamine neurons translates into changes in dopamine neurotransmission and behavioral performance in mice". We are grateful to Reviewer #1 for their encouraging statements about the manuscript.

Reviewer 2 Report

The present manuscript was well organized and interesting to readers.

In Introduction and Discussion, the authors referenced the previous papers showing the effects of inhibition or knockdown of γ-Syn. Were the behavioral and cognitive changes in mice in the present study consistent with the previous one?

Discussion might be diffuse and speculative. 

Author Response

Comments and Suggestions for Authors

The present manuscript was well organized and interesting to readers.

In Introduction and Discussion, the authors referenced the previous papers showing the effects of inhibition or knockdown of γ-Syn. Were the behavioral and cognitive changes in mice in the present study consistent with the previous one?

Discussion might be diffuse and speculative. 

Author's Reply to the Review Report (Reviewer 2)

We appreciate the time and effort that Reviewer #2 dedicated to providing feedback on our manuscript "Up-and-down γ-synuclein transcription in dopamine neurons translates into changes in dopamine neurotransmission and behavioral performance in mice". We also appreciate the positive comments and constructive criticism received.

The SNc/VTA mouse γ-synuclein overexpression model presented here leads to behavioral changes that are preferentially associated with deficits in learning and cognitive function. These changes were opposite to those previously reported using a γ-synuclein KO model. The main changes between the two models were observed in the passive avoidance and object recognition tests. Furthermore, in the Morris water maze, the data were also in the reverse direction, the higher the expression levels of γ-synuclein, the longer it took to reach the platform during each daily session and vice versa. It is important to bear in mind that although the expression of γ-synuclein was modified in both models, one of them implies the elimination of the gene during development, leaving possible adaptive changes that may affect the function of other brain circuits.

We understand the reviewer's concern/interest in the discussion of this manuscript. Since there is little information on the function of γ-synuclein in the brain and, in general, the reported data correspond to double or triple KO murine models of synuclein. Therefore, we had to focus the discussion based on these models and previous data from the group where we used a similar strategy to address the role of α-synuclein in DA neurons.

However, as the reviewer indicates, the conclusions of this study support the results obtained.
